# The Influence of (Poly)phenol Intake in Saliva Proteome: Short- and Medium-Term Effects of Apple

**DOI:** 10.3390/foods12132540

**Published:** 2023-06-29

**Authors:** Teresa Louro, Laura Carreira, Inês Caeiro, Carla Simões, Sara Ricardo-Rodrigues, Ana Elisa Rato, Fernando Capela e Silva, Henrique Luís, Pedro Moreira, Elsa Lamy

**Affiliations:** 1MED-Mediterranean Institute for Agriculture, Environment and Development & CHANGE-Global Change and Sustainability Institute, Pólo da Mitra, University of Évora, Apartado 94, 7006-554 Évora, Portugal; teresalouro@hotmail.com (T.L.); d47128@alunos.uevora.pt (L.C.); ines_caeiro_@hotmail.com (I.C.); d51413@alunos.uevora.pt (C.S.); sirr@uevora.pt (S.R.-R.); aerato@uevora.pt (A.E.R.); fcs@uevora.pt (F.C.e.S.); 2Department of Plant Science, School of Science and Technology, University of Évora, Pólo da Mitra, Apartado 94, 7002-554 Évora, Portugal; 3Department of Medical and Health Sciences, School of Health and Human Development, University of Évora, 7000-671 Évora, Portugal; 4Research Unit in Oral and Biomedical Sciences (UICOB), School of Dental Medicine and Rede de Higienistas Orais para o Desenvolvimento da Ciência (RHODes), University of Lisbon, 1649-003 Lisboa, Portugal; henrique.luis@fmd.ulisboa.pt; 5Center for Innovative Care and Health Technology (ciTechcare), Polytechnic of Leiria, 2411-901 Leiria, Portugal; 6Health School, Polytechnic Institute of Portalegre, 7300-555 Portalegre, Portugal; 7Faculty of Nutrition and Food Sciences, Porto University (FCNAUP), 4150-180 Porto, Portugal; pedromoreira@fcna.up.pt

**Keywords:** apple, phenols, saliva proteome, S-type cystatins, intake

## Abstract

The relationship between salivary proteome and dietary habits was studied in previous works, where a relationship between salivary proteins like cystatins and polyphenol/tannin levels in diet was observed. However, it remains to be elucidated if this association results from an effect of polyphenol-rich food ingestion on saliva composition. The aim of this work was to test the effects of apple intake on the saliva proteome, both in the short and medium term (after 4 days of continuous intake). By incubating saliva samples with apple phenolic-rich extract, protein bands containing α-amylase, S-type cystatins, and proline-rich proteins (PRPs) appeared in the fraction that precipitated, showing the potential of these (poly)phenols to precipitate salivary proteins. Among these, it was salivary cystatins that presented changes in their levels both in the saliva samples collected immediately after apple intake and in the ones collected after 4 days of intake of an extra amount of apple. These results support the thought that intake is reflected in the salivary proteome. The effect of a polyphenol-rich food, like the apple, on salivary cystatin levels is in line with results observed in animal models and, due to the involvement of these proteins in oral food perception, it would be interesting to explore in future studies the effect of these changes on sensory perception and acceptance of polyphenol-rich food.

## 1. Introduction

The polyphenols are among the most abundant plant phytochemical compounds in the human diet [1,2] and have been greatly investigated for their health potential [3]. The main dietary sources of polyphenols are some fruits, drinks (fruit juice, wine, tea, coffee, chocolate, and beer), vegetables, dried vegetables, and cereals [4,5,6].

Epidemiological studies suggest an association between polyphenol intake and health, what may be due to their antioxidant, anti-inflammatory, anticancer, and other bioactivity potential [2,3,7]. As such, the consumption of a high proportion of fresh fruits and vegetables in the regular diet is important to provide an adequate supply of polyphenols [6,8], with consequent health benefits. However, frequently, the intake of adequate amounts of these plant secondary metabolites is not reached, in part due to the sensory properties, namely the bitterness and/or astringency they confer to foods where they are present in high levels.

Saliva is mainly composed of water, but in its organic fraction it contains a variety of proteins. In higher proportion are the ones synthesized by salivary glands, also containing a high number of proteins with their origin in the blood. With advances in proteomics at the beginning of XXI century, the information about saliva proteins increased significantly, with more than 3000 different proteins identified in this fluid [9]. Some studies conducted on animals suggested that saliva can be deeply related with the level of polyphenols usually present in diets and with ingestive behavior through polyphenol-rich foods [10]. In some species of herbivores, the presence of salivary proteins with an affinity to high-molecular-weight polyphenols, such as tannins, was pointed out as one of the reasons for the tolerance, and intake, of these compounds [11]. Moreover, it was observed that when rodents were exposed to increased levels of tannins, the salivary glands suffered histomorphological alterations together with salivary proteome changes [12,13]. These changes were proposed as a physiological response aimed at defending individuals against potential anti-nutritive effects of these compounds. The change in rodents’ salivary proteome, in response to tannin intake, seems to influence further acceptance of bitterness and astringency [14,15].

The effect of polyphenols in human saliva has been less studied. Recently we observed a relationship between the salivary proteome and total flavanol intake [16], although it was not possible to state if the saliva proteome is the cause or consequence of polyphenol intake. The existence of an effect of polyphenol ingestion in human saliva composition is supported by some studies. One of them reported changes in the saliva proteome induced by a polyphenol-rich chocolate milk [17], and another one after a cranberry-derived polyphenol drink [18]. In both cases, variations in proline-rich proteins and cystatins were reported, in line with prior studies in animals. Moreover, by linking the known relationship between the salivary proteome and oral food perception [19,20,21] with results showing that repeated exposure to products rich in polyphenols (4 days of exposure to a high-polyphenol yerba mate/black currant beverage) leads to higher acceptance of these foods [22], the hypothesis of polyphenol intake changing oral physiology is supported. 

The objective of the present study was to test the effects of polyphenol intake in the saliva proteome, both in the short term (immediately after consumption) and the medium term (after 4 days of increased intake). For this, a counterbalanced crossover design was established, where individuals alternated a week of additional apple (variety “Bravo de Esmolfe”) consumption with a week of regular/control intake. The “Bravo de Esmolfe” is a variety of Portuguese apple, essentially coming from the Beira Alta region, more specifically originating in a small village that inspired the name of this variety—Esmolfe, located in Penalva do Castelo. This apple variety, with a greenish skin, intense and peculiar flavor, and soft pulp, has a good acceptance by the Portuguese population under study [23,24,25]. 

## 2. Materials and Methods

### 2.1. Participants

A convenience sample of twenty participants (10 from each sex), 20 to 59 years old, was randomly recruited. Only participants without self-reported and visible signs of oral (e.g., caries) or nasal health problems (e.g., loss of smell, obstruction, etc.) participated in this study. Subjects were advised not to change their normal eating habits. On the days of testing, participants were asked to arrive at the test room between 10:00 am. and 11:00 am., at least 1 h 30 m after eating breakfast and without eating or drinking anything other than water after that. Before the beginning of the study, all subjects read and signed an informed consent form. All procedures were performed according to the Declaration of Helsinki for Medical Research Involving Human Subjects and had ethical approval from the Ethical Committee of the University of Évora (GD/2746/2021).

### 2.2. Experimental Procedures

The experimental procedure had a duration of 3 weeks. The individuals were divided in two groups (Groups I and II) of 10 persons each (5 females and 5 males). During the 1st week, group I ate apple (±100 g) (variety “Bravo de Esmolfe”, from orchards in the Serra de São Mamede, Portalegre, Alentejo, Portugal) daily, in addition to their normal daily food intake. During this time, group II maintained their normal daily food intake as a control. In week 2, both groups were maintained in control conditions, without the extra apple, and in week 3, groups were exchanged and group II received the daily extra dose of apple, whereas group I remained as the control. The study design is illustrated in Figure 1.

### 2.3. Saliva Collection, Cleaning, and Protein Quantification

Saliva collection occurred daily during the five first days of week 1 and in the first five days of week 3. All participants collected saliva after arriving at the laboratory, 1 h 30 m after eating breakfast and without ingesting or drinking anything other than water after that. Before each saliva collection, individuals were asked to rinse their mouths with water to remove any food debris and “old” saliva. The first saliva collection was carried out without stimulation, by accumulating saliva in the mouth for 5 min and spitting it into a tube kept on ice. After that, the participants from the treatment group (apple) ate the apple and immediately after washed their mouths with water and produced a new saliva collection, like the one previously described. The participants from the control group ate a small amount of bread (matching the energetic content of the apple), washed their mouths with water, and collected saliva immediately after that. Bread was chosen as the control in order to have a saliva collection after the ingestion of a low-polyphenol product. Saliva was maintained on ice until it reached the laboratory (in a maximum of 1 h 30 m), where it was stored at −28 °C. Salivary flow rate (mL/min) was calculated by weighing the tubes, subtracting the weight of the empty tube, and dividing the value by the 3 min of collection. To remove mucins and cell and/or food residues, saliva samples were thawed on ice and centrifuged at 13,000× *g* for 30 min at 4 °C and supernatant was recovered. Total protein concentration was determined using the Bradford method, with Bovine Serum Albumin (BSA) as standards.

### 2.4. Extraction and Quantification of Total Phenols from Bravo de Esmolfe Apples

In order to assess the exact level of total phenols present in the apples eaten by the participants and to further use the phenolic extract for in vitro incubation with saliva samples, apple phenols were extracted. The procedure consisted of peeling the apples and separating the peel from the pulp and seeds. A total of 5 g of peel, 20 g of pulp, and 2 g of seeds were weighed. Methanol 80% (*v*/*v*) was added to the peel (15 mL), pulp (15 mL), and seeds (10 mL) fractions. All these mixtures were sonicated, on ice, for 15 min and centrifuged at 10,000× *g* rpm and 5 °C for 15 min. During this time, the tubes with the fractions were covered with aluminum foil to protect from light. Supernatants were recovered and kept at 4 °C, and pellets were extracted a second time with methanol 80% (*v*/*v*), using the same procedure described before. In this repetition, the centrifugation force was increased to 13,500× *g* rpm. Supernatants were added to the ones recovered in the first centrifugation, filtered, and the methanol evaporated. 

Phenols were measured spectrophotometrically using the Folin–Ciocalteu method, using gallic acid as reference. The phenols were measured at 750 nm and results reported as mg of gallic acid equivalents (GAEs).

This extract was aliquoted in volumes corresponding to 1 mg total phenol and frozen at −28 °C for further use in in vitro incubation with saliva samples. 

### 2.5. SDS PAGE Salivary Protein Profiles

Salivary proteins were first separated according to molecular masses using SDS-PAGE. Each saliva sample was run in duplicate, considering, for each of them, a volume corresponding to 7 µg of total protein. This volume was mixed with sample buffer and run on each lane of a 14% polyacrylamide mini-gel (Protean xi, Bio-Rad, Hercules, CA, USA) using a Laemmli buffer system, as previously described [26]. The electrophoretic run occurred at a constant voltage of 140 V, at room temperature, until the front dye reached the end of the gel. Gels were fixed for 1 h in 40% methanol/10% acetic acid, followed by staining for 2 h with 2% Coomassie Brilliant Blue (CBB) R-250 and destaining in several washings of 10% acetic acid. The gel images were acquired using a scanning Molecular Dynamics densitometer with internal calibration and LabScan software (GE Healthcare, Chicago, IL, USA), with images analyzed using the GelAnalyzer software (GelAnalyzer 2010a by Istvan Lazar). Molecular masses were determined in accordance with molecular mass standards (Bio-Rad Precision Plus Protein Dual Colour 161–0394) run with protein samples.

### 2.6. In Vitro Analysis of Saliva*Polyphenol Extract Interactions

To assess the salivary proteins with capacity for interacting with apple polyphenols, saliva samples from all participants (N = 20), collected before and after apple ingestion, were incubated with the polyphenolic extract previously obtained, simulating mouth conditions. A total of 1 mg polyphenolic extract was added to an equal volume of saliva in a polyethylene tube. For control, the same volume of saliva was mixed with an equal volume of distilled water, instead of polyphenolic extract. Samples were incubated for 30 min at 37 °C to allow for salivary proteins*polyphenol complexation, after which the mixture was centrifuged at 15,000× *g*, at 4 °C, for 15 min. The supernatant and precipitate were collected in different tubes and proteins included in each were subsequently separated using SDS-PAGE, as previously described in Section 2.5. 

### 2.7. Two-Dimensional Electrophoretic (2DE) Salivary Protein Profiles

With the aim of going deep into the potential changes that apple polyphenol intake could induce in the salivary proteome, a sub-sample of the participants, consisting of six individuals (3 females and 3 males randomly selected), was studied for the two electrophoretic profiles of their saliva samples. To minimize technical errors, each sample was run in duplicate. At the end, a total of 48 2DE gels (6 individuals × 4 collection moments × 2 technical repetitions) were analyzed.

A volume of each saliva sample, corresponding to a total of 175 µg of total protein, was concentrated and de-salted using ultracentrifugation membranes with 3 kDa cut-off (centricon), at 13,000× *g* and 4 °C. The centrifugation time was necessary to recover a volume of sample lower than 50 µL. The volume of the concentrated saliva sample was mixed with rehydration buffer [7 M urea, 2 M thiourea, 4% (*w*/*v*) CHAPS, 2% (*v*/*v*), 60 mM DTT and traces of bromophenol blue] + 2.5 µL IPG buffer to achieve a final volume of 250 μL and loaded onto 13 cm pH 3–10 NL IPG strips (GE Healthcare) by passive in-gel rehydration overnight in a Multiphor Reswelling Tray (GE Healthcare). Strips were focused for a total of 30 kVh at 18 °C, using a Multiphor II isoelectric focusing system (GE Healthcare). Focused strips were equilibrated in two steps of 15 min each [50 mM Tris–HCl, pH 8.8; 6 M urea; 30% (*v*/*v*) glycerol and 2% (*w*/*v*) SDS], with the addition of 1% (*w*/*v*) DTT and 65 mM iodoacetamide in the first and second steps, respectively. After equilibration, the strips were applied in the top of SDS-PAGE gel 12% acrylamide and run at 150 V constant voltage in a Protean II xi cell (Bio-Rad). Gels were stained with CBB-G250, as indicated for SDS-PAGE gels. Gel images were acquired using a gel scanner (Epson) and Labscan software. ImageMaster 2D Platinum v7 software was used to analyze gel images. Spot editing and the match were performed automatically and corrected manually. Spot volume was normalized to the total spot volume. 

### 2.8. Statistical Analysis

Descriptive statistics were performed, and data normal distribution and homoscedasticity were tested through Shapiro–Wilk and Levene tests, respectively. 

To evaluate either short-term or medium-term effects of apple intake on the different salivary parameters evaluated (flow rate, total protein concentration, % volume of each protein band, and % volume of each spot), two-way ANOVA repeated measures analysis was used. In both cases, period was considered as the within-subjects factor (with two levels: before and after treatment) and treatment (apple and control) was considered as the between-subjects factor. 

Besides this analysis, for each of the separate treatments, the periods before and after apple intake were compared using Wilcoxon signed rank test.

All these statistical procedures were performed for saliva total protein concentration and percentage volume of each SDS-PAGE band. Statistical analysis was performed using SPSS v. 24. For all statistical analysis, significance level was set at 5%.

## 3. Results

### 3.1. Effects of Apple Intake on Salivary Total Protein and SDS-PAGE Profile

#### 3.1.1. Short-Term Effects

Flow rate significantly changed after treatment (*p* = 0.010) and a significant interaction between treatment and period was observed (*p* = 0.017), which increased only after apple intake. Although presenting a tendency for increasing, no statistically significant changes in total protein concentration after apple intake were observed.

After protein separation, by SDS-PAGE, a total of 17 protein bands were considered for analysis (Figure 2). The correspondence between bands and salivary proteins was made according to the identifications in previous studies, where mass spectrometry was used for protein identification in equivalent SDS-PAGE salivary profiles [19,20], including the identification of proline-rich proteins (PRPs) in the bands stained pink with the staining procedure used [27,28].

Some protein bands responded differently to the apple when compared to the control (Table 1). Interaction between treatment and period was observed for band *d* (*p* = 0.012), band *f* (*p* = 0.003), band *g* (*p* = 0.003), and band *r* (*p* = 0.044). 

For bands *d* (predicted to contain albumin) and *f* (predicted to contain amylase), increases were observed only after apple intake, whereas for bands *o* (predicted to contain PRPs) and *w* (predicted to contain cystatins), significant increases occurred after intake of both type of foods (apple and bread—control). For band *r* (predicted to contain PRPs), no significant difference was observed after apple intake, although the significant interaction shows that eating bread or apple has a different effect, with apple causing an increasing direction. 

#### 3.1.2. Medium-Term Effects

When saliva was collected after 4 days of added consumption of the apple, no statistically significant differences in salivary parameters were observed. However, for band w, the interaction between treatment (apple and bread) and period (before and after) is significant, showing a different effect of the apple when compared to the control food (Table 2). In this case, there was a tendency for higher levels in band w after apple consumption (*p* = 0.089). 

### 3.2. Interaction between Salivary Proteins and Apple Phenolic Extract

Folin–Ciocalteu quantification of total phenol amount of the apples used in this study resulted in a mean amount of 106.04 ± 25.54 mg GAE (gallic acid equivalent) per 100 g fresh weight. 

In order to assess the affinity of salivary proteins to precipitate phenols/polyphenols of the apple, saliva samples were incubated with the apple phenolic extract, with precipitated and non-precipitated fractions recovered for electrophoretic analysis. From this, it was possible to observe that several salivary proteins were present in the precipitate at higher levels when incubation was carried out with the phenolic extract, as compared to when incubation was carried out with water, namely bands e, f, h, i, j, n, o, and w (white arrows in Figure 3), meaning that proteins like α-amylase, CA-VI + zinc-α2-glycoprotein + SPLUNC, and S-type cystatins bind and precipitate with these phenols to a certain degree, as well as PRPs (according to the pink staining observed for bands n and o. 

When looking for the supernatant fraction, only three protein bands were significantly decreased after incubation with the phenolic extract (red arrow heads in Figure 3), as compared to incubation with water, namely l, m, and n, the latter being a band that was simultaneously observed to increase in precipitate and to decrease in supernatant.

### 3.3. Effect of Apple Intake on Salivary 2-DE Profile

Variations in salivary 2DE profiles were observed both for short- and medium-term periods, where the levels of different spots changed, as compared to the period before apple ingestion. The significant interactions between time and treatment observed for most of the spots show that for some proteins, the variation induced by intake was not the same for the apple as for the bread (Table 3). In the short term, 23 spots changed after apple intake, 9 of which increased and 14 of which decreased. Protein identification was obtained from the results of previous studies (referred to in Table 3). Proteins such as cystatins, carbonic anhydrase VI, prolactin inducible protein, and SPLUNC appear to increase in relative amounts immediately after apple intake, whereas proteins like zinc-α-2-glycoprotein, IgK chain C region, and α-amylase decreased. Besides these 23 spots, the other 5 did not differ significantly between the period before and after intake, but a significant interaction was observed, indicating that the direction of variation is different with apple or control (bread) intake.

In the medium-term additional apple intake, 16 spots changed, 5 of which were observed to increase and 11 of which decreased. The increases (or tendencies to increase) were the levels of proteins such as cystatins and carbonic-anhydrase VI, while the decreases were in zinc-α-2-glycoprotein. For albumin and amylase identified spots, there were forms increased and forms decreased. 

Figure 4 presents the location of the spots for which statistically significant changes (or significant interactions between period and treatment) were observed. 

## 4. Discussion

The effect of food intake in the saliva proteome has been reported in different studies [13,33]. Tannins are probably the compounds present in vegetable foods more associated with direct effects at the salivary protein level, since these polyphenolic compounds have a particular affinity for some types of salivary proteins, binding them. For some animal species, the level of dietary tannins was associated with the presence/absence of tannin-binding proteins, among which proline-rich proteins [34] and intake of tannin-enriched diets result in changes in salivary protein composition [12,35,36]. In humans, this direct effect of tannins on saliva protein composition has been less explored, but an association between the usual levels of total polyphenol intake and the salivary proteome has been observed [16,17,37]. Since salivary proteins also participate in food oral perception [38,39], it is difficult to be sure about a cause–effect direction, i.e., whether it is salivary protein composition that influences polyphenol intake habits through its influence on sensory perception, or whether it is polyphenol intake habits that influence saliva composition. To assess the effect of short-term and medium-term intake of polyphenols, we analyzed the salivary proteome of samples collected immediately after apple intake and after 4 days of additional apple intake. 

Apple was chosen for being a food rich in (poly)phenols, easily eaten by most people at any place. A variety characteristic of the region of Portugal, where the study took place, was chosen, namely the “*bravo de esmolfe*” apple. The level of total phenols was confirmed analytically, attesting that the participants ingested extra levels of phenols. The main phenolics of this variety of apple are epicatechin and procyanidins [23], the latter having a particularly high capacity to bind proteins, being responsible for astringency [40]. This capacity for binding salivary proteins for the apples used in the study was confirmed when saliva samples were incubated with apple phenolic extract and a precipitate was formed. The salivary proteins α-amylase, S-type cystatins, and PRPs were observed to be the ones with higher affinity for precipitate of the (poly)phenols present in the apple. These, together with histatins, are the salivary proteins usually said to have the highest affinity to bind and precipitate polyphenols (tannins) [41]. These proteins were previously observed in the precipitate fraction of saliva samples incubated with polyphenols from green tea [42] and wine [43], with PRPs being the salivary proteins more widely accepted to precipitate tannin polyphenols [44]. The presence of these proteins in the precipitate that resulted from incubation of saliva with apple phenolic extract reinforces the high affinity of these proteins to bind polyphenols from different origins. Although astringency was not evaluated in the apples that were consumed, in this study, it is possible to hypothesize a certain level of this sensation, in which these proteins could be involved. In the present study, the band with molecular mass of the band(s) containing CA-VI + zinc-α2-glycoprotein + SPLUNC (identifications obtained in previous studies: [19,21]) was also observed in the precipitate fraction resulting from the incubation of saliva with apple phenolic extract. CA-VI has been related to taste sensitivity, mainly bitterness sensitivity [19,45], but not usually reported as a tannin-binding protein. Since this band also stains pink (at a variable degree for different individuals), it is possible that, in the precipitate, it is mainly constituted by PRPs.

Apple intake induced some changes in the salivary proteome, both immediately after ingestion (short term) and after some days of ingestion of extra amounts (medium term). This reinforces the idea that eating habits may influence saliva protein composition, suggesting this fluid as having potential as a source of ingestion biomarkers. With these results it is possible to suggest that the association between salivary protein profile and the levels of polyphenols present in the usual diet, observed in our previous studies [16,37], is at least in part due to daily food habits. 

The short-term plasticity of saliva in response to food or food stimuli is not a surprise, since it was previously observed in studies from our lab [33,46] and from other authors [47]. Looking for SDS-PAGE profiles, our participants increased the relative levels of the bands expected to contain albumin, α-amylase, PRPs, and cystatins, which, with the exception of albumin, are all salivary proteins observed to bind and precipitate apple polyphenols, as discussed above. When looking for the 2DE profiles, changes were not observed for PRP spots, which may be due to the poor separation capacity of 2DE for PRPs, particularly at the level of basic PRPs, which are the ones more frequently reported to be associated to polyphenol/tannin intake, which act as a defense mechanism against potential anti-nutritional effects of these molecules [48,49], and which have isoelectric points at the extreme of the gel. In addition, in 2DE profiles it was not possible to see the increase in protein spots of albumin or amylase, as was observed in SDS-PAGE. In fact, the few protein spots of these proteins observed to change changed in the opposite direction. The lower number of individuals whose saliva was analyzed by 2DE may explain these different results. 

Concerning salivary cystatins, increases were observed both for the band, in SDS-PAGE gels, and spots, in 2DE gels. It was even more interesting to see that this increase was maintained when saliva was collected after some days of eating the extra polyphenol dose from apples. The short-term increase in salivary cystatin levels after polyphenol-rich foods is in line with previous studies, showing increases in this protein after intake of cranberry juice [18] or apple [33], and with studies relating these salivary proteins with different blood levels of caffeine [50]. This apparent increase in salivary cystatins as a response of dietary phenol levels is in line with our previous studies showing a degree of association between these salivary proteins and food patterns, whereas patterns characterized by higher amounts of polyphenol-rich products have positive associations with relative amounts of S-type salivary cystatins [16,37]. 

The medium-term changes in the salivary proteome reinforces the idea that the type of eating habits will affect saliva composition. This is not a new hypothesis, since changing diet for several days had already been observed to induce salivary changes in animals [13,36] and, in the case of humans, Santos and colleagues related salivary amylase variations between individuals to starch levels in diets [51]. The increases in salivary cystatin levels are particularly interesting both in terms of potential effects in acceptance and/or nutrient bioavailability. Studies using rodents as animal models showed that increases in salivary proteins, S-type cystatins among them, resulting from tannin intake, have as a consequence a higher acceptance and ingestion of bitter solutions [14,15]. Nutritional consequences can also be expected. Despite the limited number of studies, Delimont and colleagues observed that in diets with phytic acid, salivary cystatin SN concentrations were positively correlated with iron bioavailability [52].

The observation that diet influences salivary protein composition does not mean that the saliva composition will not influence food acceptance and choices. In fact, it is known that repeated exposure to foods will affect the acceptance of the sensory characteristics of those foods [22]. The familiarity with sensory stimuli and the link with post-ingestive effects is known to relate to this increased acceptance, but it is not to discard the hypothesis of the changes in saliva composition as also being a part of the change. However, this hypothesis needs to be tested in future studies.

The present study has some limitations that need to be considered. First, only one type of polyphenol-rich food was studied, and conclusions may be not totally assumed for all the foods containing these plant secondary compounds; second, the total amount of polyphenols present in the participants’ total diet during the 5 days of the experiment was not accessed through food diaries or other types of daily reports. Due to the difficulty in having major control and the impossibility to force individuals to eat according our recommendations, we needed to work within this limitation and assume that each participant kept their usual habits, the only change being the introduction of the 300 g of apple. The variations in the medium term, in line with the short term and other studies, as previously discussed, make us assume that participants accomplished the task as they attested. It is also important to highlight that a convenience sample was studied consisting of 20 individuals, making it necessary to be careful in extrapolations of the results to the general population. Further studies in a higher number of individuals from populations in which the global diet can be controlled (e.g., prisons or intern institutions) can be of interest to confirm these results.

## 5. Conclusions

This study shows that the intake of a polyphenol-rich food induces changes in the salivary protein profile, not only immediately, or some minutes after intake, but also if that intake is kept up for some days. With the experimental design and the proteomic approach used, it is mainly for salivary cystatins that the effect was simultaneously seen in the short and medium term. Being that these proteins are involved in the perception of astringency and bitterness present in vegetable-based products, the effect of dietary polyphenols in saliva may also have consequences in further sensory perception, which may influence acceptance and preference.

## Figures and Tables

**Figure 1 foods-12-02540-f001:**
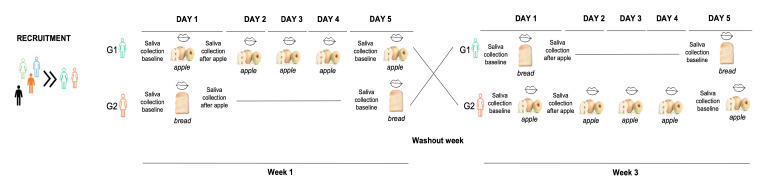
Schematic representation of experimental design.

**Figure 2 foods-12-02540-f002:**
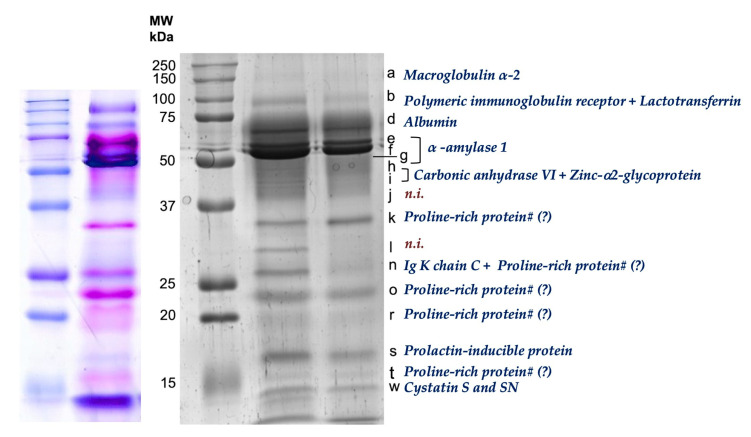
Example of salivary SDS-PAGE protein profiles. Letters on the right side correspond to each of the protein bands considered for analysis; identifications are based on previous studies (e.g., [19,20]). In the left gel image it is possible to observe pink bands, considered as containing PRPs.

**Figure 3 foods-12-02540-f003:**
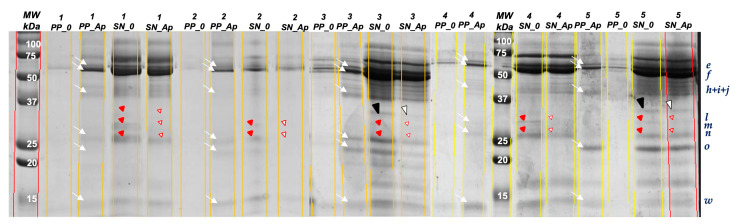
SDS-PAGE profiles of supernatant and precipitate fractions of saliva samples incubated with apple phenolic extract (white arrows—bands present in the precipitate fraction; full or open red arrow heads—bands decreased in the supernatant fraction; full or open black arrow head—band observed to decrease from the supernatant in several samples, but not appearing in the precipitate fraction).

**Figure 4 foods-12-02540-f004:**
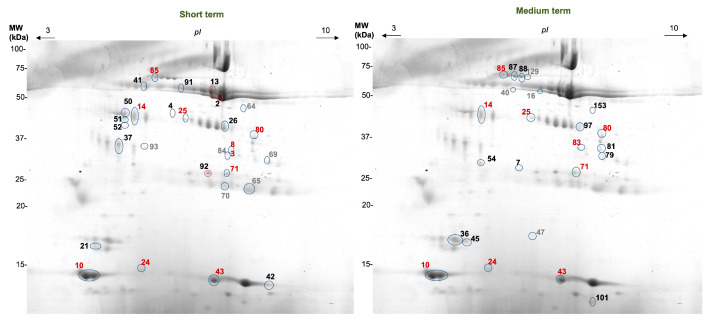
Representative image of 2DE salivary protein profiles. Numbers represent the spots for which apple-induced significant changes or treatment period significant interactions occurred. (red—similar changes after short- and medium-term treatment).

**Table 1 foods-12-02540-t001:** Short-term variations in salivary parameters (mean ± standard deviation) with apple intake and differences between these and the ones induced by the intake of bread as control food.

Salivary Parameter	Apple			Control (Bread)			Treat * Period (*p*-Value)
	Before	After	*p*-Value	Before	After	*p*-Value	
Flow rate	0.53 ± 0.28	0.71 ± 0.32	0.001 *	0.49 ± 0.30	0.50 ± 0.31	0.879	0.017 *
Total protein concentration	675.9 ± 336.6	982.9 ± 377.7	0.062	370.4 ± 72.4	376.1 ± 112.5	0.878	0.111
Band a	4.41 ± 1.78	3.40 ± 1.28	0.155	4.64 ± 1.53	6.24 ± 2.75	0.418	0.166
Band b	6.88 ± 2.26	6.19 ± 2.27	0.445	8.75 ± 1.83	8.07 ± 1.44	0.491	0.992
Band d	8.71 ± 3.52	12.83 ± 5.33	0.008 *	11.25 ± 4.43	9.44 ± 4.62	0.338	0.012 *
Band e	8.29 ± 2.30	8.89 ± 3.30	0.570	11.77 ± 5.80	8.09 ± 1.81	0.103	0.051
Band f	8.25 ± 2.10	12.98 ± 3.63	0.001 *	11.26 ± 3.83	9.77 ± 3.38	0.371	0.003 *
Band g	3.63 ± 1.96	5.23 ± 3.58	0.261	7.57 ± 4.15	3.63 ± 3.69	0.126	0.035 *
Band h	4.22 ± 2.30	3.46 ± 2.94	0.576	3.10 ± 1.54	3.48 ± 1.51	0.552	0.503
Band i	3.60 ± 1.66	3.02 ± 1.15	0.496	3.98 ± 1.06	3.39 ± 1.55	0.180	0.990
Band j	3.88 ± 1.37	4.76 ± 2.32	0.208	4.64 ± 1.33	6.42 ± 0.48	0.129	0.471
Band k	4.68 ± 1.76	8.06 ± 4.25	0.335	3.53 ± 1.59	4.91 ± 2.05	0.562	0.626
Band l	4.46 ± 1.60	3.20 ± 0.58	0.140	3.43 ± 0.25	3.91 ± 0.66	0.346	0.194
Band n	8.55 ± 2.30	8.09 ± 2.99	0.615	9.67 ± 2.38	6.78 ± 2.07	0.032 *	0.112
Band o	6.05 ± 1.59	8.04 ± 2.11	0.012 *	6.84 ± 1.23	9.19 ± 1.89	0.006 *	0.718
Band r	4.87 ± 1.16	5.37 ± 1.36	0.469	5.26 ± 1.21	3.94 ± 0.77	0.041 *	0.044 *
Band s	6.10 ± 3.18	5.90 ± 2.18	0.881	4.36 ± 0.50	4.70 ± 0.06	0.471	0.837
Band t	5.27 ± 3.30	4.14 ± 2.18	0.479	2.18 ± 0.25	2.27 ± 0.76	0.916	0.691
Band w	6.83 ± 2.36	9.46 ± 2.79	0.004 *	10.97 ± 2.83	12.97 ± 2.06	0.014 *	0.568

* Significant for *p* < 0.05; grey shadows highlight statistically significant results.

**Table 2 foods-12-02540-t002:** Medium-term variations in salivary parameters (mean ± standard deviation) with apple intake and differences between these and the ones induced by the intake of bread as control food.

Salivary Parameter	Apple			Control (Bread)			Treat * Period (*p*-Value)
	Before	After	*p*-Value	Before	After	*p*-Value	
Flow rate	0.61 ± 0.23	0.59 ± 0.26	0.778	0.48 ± 0.13	0.57 ± 0.35	0.479	0.400
Total protein concentration	703.0 ± 332.6	555.0 ± 363.2	0.111	555.3 ± 237.3	708.4 ± 389.7	0.313	0.065
Band a	3.72 ± 0.97	4.73 ± 1.19	0.158	4.78 ± 2.38	5.17 ± 0.69	0.704	0.609
Band b	6.21 ± 1.89	6.46 ± 1.47	0.598	8.22 ± 2.58	7.23 ± 1.28	0.489	0.295
Band d	8.61 ± 3.92	7.81 ± 2.20	0.577	8.52 ± 2.98	6.37 ± 1.91	0.219	0.551
Band e	7.85 ± 2.07	8.01 ± 1.79	0.857	11.24 ± 3.76	9.69 ± 3.53	0.525	0.419
Band f	8.09 ± 1.64	9.12 ± 2.82	0.311	8.24 ± 3.93	9.23 ± 2.54	0.548	0.982
Band g	3.48 ± 1.54	3.48 ± 1.03	0.997	2.57 ± 0.59	3.40 ± 0.82	0.052	0.395
Band h	4.09 ± 2.34	4.57 ± 3.63	0.762	2.94 ± 0.99	2.44 ± 0.46	0.337	0.621
Band i	3.47 ± 1.75	5.52 ± 3.68	0.215	3.62 ± 1.16	3.20 ± 1.15	0.675	0.281
Band j	3.90 ± 1.05	4.12 ± 1.75	0.777	3.27 ± 1.09	3.74 ± 0.69	0.604	0.818
Band k	4.94 ± 1.55	4.26 ± 1.16	0.324	4.97 ± 1.38	2.98 ± 0.84	0.059	0.228
Band l	4.43 ± 1.50	3.79 ± 1.87	0.286	3.91 ± 1.53	3.08 ± 1.22	0.639	0.883
Band n	8.68 ± 2.01	8.26 ± 2.91	0.724	9.06 ± 1.24	8.57 ± 1.40	0.472	0.966
Band o	6.59 ± 2.07	6.91 ± 2.28	0.634	6.93 ± 3.08	7.00 ± 1.19	0.963	0.861
Band r	5.24 ± 1.31	4.72 ± 1.74	0.571	5.30 ± 1.59	5.30 ± 1.47	0.996	0.697
Band s	5.48 ± 1.93	3.88 ± 1.85	0.314	4.39 ± 1.09	4.69 ± 1.57	0.681	0.295
Band t	4.20 ± 1.69	4.48 ± 1.86	0.816	3.03 ± 1.33	3.74 ± 0.43	0.289	0.799
Band w	6.39 ± 2.15	7.40 ± 2.17	0.089	9.13 ± 2.98	7.14 ± 2.15	0.239	0.029 *

* Significant for *p* < 0.05; grey shadows highlight statistically significant results.

**Table 3 foods-12-02540-t003:** Variations in salivary 2-DE protein spots with apple intake and differences between these and the ones induced by the intake of bread as control food.

Spot	Apple		Control (Bread)		Interaction T * *p* (*p*-Value)	Protein ID	Refs
	Change	*p*-Value	Change	*p*-Value			
*Short term*							
2	↓	0.028 *	---	0.173	0.017 *	α-Amylase	[29,30,31]
4	↑	0.043 *	↑	0.046 *	0.023 *	n.i.	
10	↑	0.046 *	↑	0.028 *	0.609	Cystatins S,SN	[21,29]
13	---	0.753	↑	0.028 *	0.018 *	α-Amylase	[20,29,30,31]
14	↓	0.028 *	↑	0.046 *	0.0005 *	Zinc α-2 glycoprotein	[30,31]
21	↑	0.028 *	---	0.249	0.667	Prolactin inducible protein (PIP)	[30]
24	↓	0.043 *	---	0.249	0.038 *	n.i	
25	↓	0.046 *	---	0.753	0.165	n.i.	
26	↑	0.028 *	---	0.116	0.782	Carbonic anhydrase VI (CA-VI)	[30]
37	↑	0.043 *	↑	0.028 *	0.146	SPLUNC	[31]
41	↓	0.028 *	↓	0.046 *	0.244	Ig α chain C	[19]
42	↓	0.075	↑	0.028 *	0.0005 *	n.i	
43	↑	0.028 *	↑	0.043 *	0.052	Cystatins S,SN	[19,20,21]
50	↑	0.028 *	---	0.686	0.001 *	Actin cytoplasmic 1	[21,29,30,31]
51	↓	0.068	↑	0.028 *	0.0005 *	Zinc α-2 glycoprotein	[30]
52	↓	0.028 *	↑	0.028 *	0.0005 *	Zinc α-2 glycoprotein	[30]
64	---	0.116	---	0.345	0.034 *	α-Enolase	[31]
65	↓	0.028 *	↓	0.028 *	0.316	IgK chain C region	[29,31]
69	↓	0.028 *	---	0.249	0.002 *	n.i.	
70	↓	0.046 *	---	0.116	0.280	IgK chain C region	[29,31]
71	↑	0.028 *	↓	0.028 *	0.0005 *	IgK chain C region	[29,31]
80	↓	0.028 *	---	0.917	0.0005 *	n.i.	
83	↓	0.046 *	---	0.225	0.009 *	n.i	
84	↓	0.043 *	↓	0.028 *	0.256	n.i	
85	---	0.500	↑	0.028 *	0.022 *	Albumin	[29,30,31]
91	↓	0.046 *	---	0.600	0.120	Ig α chain C	[29]
92	↑	0.028 *	↓	0.028 *	0.0005 *	IgK chain C region	[19,29,31]
93	↓	0.028 *	↓	0.046 *	0.010 *	Zinc α-2 glycoprotein	[31]
*Medium term*							
7	↑	0.043 *	---	0.109	0.002 *	IgK chain C region	[32]
10	↑	0.173	↓	0.043 *	0.009 *	Cystatins S,SN	[21,29]
14	↓	0.028 *	---	0.144	0.320	Zinc α-2 glycoprotein	[30,31]
16	↓	0.028 *	---	0.686	0.002 *	α-Amylase	[21,30,31]
24	↑	0.173	---	0.109	0.047 *	n.i	
25	↓	0.028 *	---	0.686	0.017 *	n.i	
36	↑	0.075	↓	0.043 *	0.008 *	Prolactin inducible protein (PIP)	[21,30]
40	↑	0.043 *	---	0.500	0.097	α-Amylase/Ig α chain C	[19,30,31]
43	---	0.249	↓	0.043 *	0.003 *	Cystatins S,SN	[19,20,21]
45	↓	0.043 *	---	0.893	0.121	Prolactin inducible protein (PIP)	[31]
47	↓	0.028 *	---	0.225	0.005 *	n.i	
54	↑	0.075	---	0.273	0.041 *	SPLUNC	[31]
71	↓	0.046 *	---	0.893	0.064	IgK chain C region	[21]
79	↑	0.028 *	---	0.686	0.051	n.i.	
80	↓	0.028 *	---	0.180	0.006 *	n.i.	
81	↑	0.068	---	0.138	0.008 *	n.i.	
83	↓	0.028 *	---	0.465	0.032 *	n.i.	
85	↑	0.028 *	---	0.225	0.036 *	Albumin	[29,30,31]
87	↓	0.028 *	---	0.080	0.434	Albumin	
88	↓	0.028 *	---	0.345	0.302	Albumin	
97	↑	0.043 *	---	0.500	0.017 *	Carbonic anhydrase VI (CA-VI)	[31]
101	---	0.463	↓	0.043 *	0.024 *	n.i.	
129	↓	0.028 *	---	0.080	0.004 *	Albumin	[29,30,31]

* Significant for *p* < 0.05; ↑ and ↓ represent increase and decrease, respectively, between periods before and after intake; --- means the change was not statistically significant; grey arrows (for apple treatment) indicate a tendency of change, although not statistically significant; cells in grey indicate the spots simultaneously changed in short and medium term; cells in blue indicate the spots that change in both periods, but in opposite direction. Ref indicates the studies where the proteins corresponding to the spot were identified. n.i.—not identified.

## Data Availability

The data presented in this study are available on request from the corresponding author. All data relevant to the study are included in the article and access to raw data would be provided upon request.

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
