# Peer review of "The Influence of (Poly)phenol Intake in Saliva Proteome: Short- and Medium-Term Effects of Apple"

_foods, 2023, doi:10.3390/foods12132540_

Round 1
Reviewer 1 Report
It is important to start the introduction with information about the saliva proteome and then explain factors that can modify it.
It is a thorough and detailed work, however (probably because of what the authors mention in their limitations, coupled with the small sample size) the results obtained in vivo are poor. Not so the in vitro results, which are very rich. Why force the in vivo results and not take more advantage of those obtained in vitro? I would suggest to modify the point of view of the work until you can gather a larger number of volunteers and obtain stronger results.
The results obtained on the proteomic changes in saliva in the short and medium term after apple consumption are mostly not significant. You do obtain differences when saliva is incubated with apple phenolic-rich extracts. This is not so clear in the discussion or in the abstract. 4 days seems to be too short time to observe significant differences in such a small study group.
Unify whether or not to leave a space between the number and the unit of measurement (see journal regulations). E.g.: 15mL or 15 mL; 20% or 20%; 106.04±25.54 or 106.04 ± 25.54.
Word file with suggestions and corrections is attached.

Some corrections were made in the text, others are indicated. A revision would be desirable.
Reviewer 2 Report
The manuscript discusses the relationship between dietary habits and the salivary proteome. Previous studies have shown a correlation between salivary proteins and polyphenol/tannin levels in the diet. The aim of this study was to investigate the effects of apple intake on the salivary proteome in both the short and medium term. The results showed that apple consumption led to changes in the salivary proteome, particularly an increase in salivary cystatins. The study suggests that polyphenols in apples have the potential to precipitate salivary proteins and affect the composition of the salivary proteome. Future studies could explore the effect of these changes on sensory perception and acceptance of polyphenol-rich foods.
Overall, the article provides interesting insights into the potential benefits of consuming polyphenols in apples and their effects on the salivary proteome. However, it should be noted that the study was conducted on a small group of volunteers, and further research is necessary to confirm these findings and to investigate the long-term effects of polyphenol intake on the proteome of saliva.
Author Response
We would like to thank the reviewer for the positive comments to our work. We agree with the limitation of the samples size and included this, as limitation, at the end of discussion section.
Some minor corrections were performed at abstract and and discussion level, to highlight the results obtained by in vitro, which we believe to improve the manuscript.